# Improved Conductivity in Gellan Gum and Montmorillonite Nanocomposites Electrolytes

**DOI:** 10.3390/molecules27248721

**Published:** 2022-12-09

**Authors:** Willian Robert Caliman, Franciani Cassia Sentanin, Rodrigo Cesar Sabadini, Jose Pedro Donoso, Claudio Jose Magon, Agnieszka Pawlicka

**Affiliations:** 1Instituto de Química de São Carlos, Universidade de São Paulo, Av. Trabalhador Saocarlense 400, São Carlos 13566-590, SP, Brazil; 2Instituto de Física de São Carlos, Universidade de São Paulo, Av. Trabalhador Saocarlense 400, São Carlos 13566-590, SP, Brazil

**Keywords:** gellan gum, ionic conductivity, nanocomposite polymer electrolytes

## Abstract

Nanocomposite polymer electrolytes (NPEs) were obtained using gellan gum (GG) and 1 to 40 wt.% of montmorillonite (Na^+^SYN-1) clay. The NPEs were crosslinked with formaldehyde, plasticized with glycerol, and contained LiClO_4_. The samples were characterized by impedance spectroscopy, thermal analyses (TGA and DSC), UV-vis transmittance and reflectance, X-ray diffraction (XRD), and continuous-wave electron paramagnetic resonance (CW-EPR). The NPEs of GG and 40 wt.% LiClO_4_ showed the highest conductivity of 2.14 × 10^−6^ and 3.10 × 10^−4^ S/cm at 30 and 80 °C, respectively. The samples with 10 wt.% Na^+^SYN-1 had a conductivity of 1.86 × 10^−5^ and 3.74 × 10^−4^ S/cm at 30 and 80 °C, respectively. TGA analyses revealed that the samples are thermally stable up to 190 °C and this did not change with clay addition. The transparency of the samples decreased with the increase in the clay content and at the same time their reflectance increased. Finally, CW-EPR was performed to identify the coordination environment of Cu^2+^ ions in the GG NPEs. The samples doped with the lowest copper concentration exhibit the typical EPR spectra due to isolated Cu^2+^ ions in axially distorted sites. At high concentrations, the spectra become isotropic because of dipolar and exchange magnetic effects. In summary, GG/clay NPEs presented good ionic conductivity results, which qualifies them for electrochemical device applications.

## 1. Introduction

Since the mid-1980s nanocomposites produced from the intercalation of nanoscale clay into polymers have attracted great interest in research and industry because they feature improvements in several properties when compared to samples without fillers [1]. Nanocomposites are hybrids made by dispersion at low concentrations of nanoscale fiber or particle reinforcements in polymeric matrices [2]. Clays are abundant, low-cost, raw materials, which are also compatible with polymers [3]. The application of clays in polymer nanocomposites results in materials with exfoliated or intercalated structures because of the layered structure of clay [3]. Many groups are developing a growing number of products with these nanocomposites such as biosensors [4], solid-state batteries [5], electrochromic devices [1], and hydrogels [6]. In the studies on the ionic conducting nanocomposites, polymer-ion, polymer-clay and ion-clay interactions play an important role in ionic conductivity. For example, looking at the ion–clay interaction it is observed that the negative net charge on the nanoparticle surface helps the polymer heteroatoms to dissociate the lithium salts, leading to an increase in charge carriers. On the other hand, in the interaction between polymer and clay, the lamellae may decrease the crystallinity of the polymer and increase its amorphousness, which improves the ionic transport through the polymer-clay matrix [7].

Chen and Chang [8] demonstrated that the addition of modified montmorillonite clay (MMT) to a poly(ethylene oxide) (PEO) matrix increased the samples’ ionic conductivity up to 16 times. Three types of complexes were formed between the polymer, clay lamellae and Li^+^. Additionally, the FTIR spectra showed that the presence of clay increased salt dissolution, contributing to the increase in free ion concentration. Chen et al. [9] showed that increasing clay concentration modified by alkylammonium salts and surfactants decreased the glass transition temperature of PEO-based solid polymer electrolytes (SPE). In this case, there was also the formation of three distinct types of Li^+^ complexes in the system [9]. Walls et al. [10] compared the morphology, rheology and ion transport of nanocomposite gel electrolytes of PEO containing silica or hectorite clay. Although there was an interaction between charge and ion only in the second case, the conductivity exceeding 10^−3^ S/cm of the silica-containing electrolyte was higher than that with clay. Wang et al. [11] indicated that the ionic conductivity of poly(vinylidene fluoride-hexafluoropropylene) nanocomposite polymeric electrolytes (PVDF-HFP) and alkylammonium salt-modified clay can increase from 10^−6^ to 10^−3^ S/cm with the addition of plasticizers. Organically modified Cloisite 30B clay improved the ionic conductivity of PEO nanocomposite electrolytes containing NaClO_4_. This was due to the exfoliated structure, unlike when applying laponite clay, which interacted with the polymer producing an intercalated structure and resulting in conductivity at the same level as pristine PEO [12]. Poly(methyl methacrylate) (PMMA)—containing plasticizer, organically modified MMT clay with octadecyl dimethyl betaine surfactant and LiClO_4_—with 1.5 wt.% of clay had a superior conductivity compared to the polymer electrolyte without clay and it was applied as NPE to batteries [13]. In 2007, the urethane acrylate non-ionomer (UAN) precursor chain was mixed with several nonpolar monomers and Li^+^ salts for the manufacture of solvent-free SPE films by Kim et al. [14]. SPE-containing butyl acrylate showed the highest values of ionic conductivity. However, when Cloisite^®^ clay was added to styrene samples, they demonstrated improved ionic conductivity from 10^−5^ to 10^−4^ S/cm. Salavagione et al. [15] prepared poly(o-methyl aniline) nanocomposites containing maghnite—Algerian montmorillonite clay—containing intercalated copper or strong acid cation between the clay layers. X-ray diffraction (XRD) and transmission electron microscopy (TEM) measurements showed that the sample containing Cu cations resulted in an intercalated structure while the sample with strong acid presented an exfoliated structure. Cyclic voltammetry revealed that the polymer extracted from the clay with Cu had a good electrochemical response, while that derived from strong acid clay showed no redox process. Protonic montmorillonite was added to Nafion by Alonso et al. [16] for the study of materials to be applied in fuel cells. Despite the intercalated/exfoliated structure, improved thermomechanical resistance and reduced methanol permeability of nanocomposite compared to pristine material, there was a decrease in proton conductivity with increasing H^+^MMT concentration. Ionic transport of polyacrylonitrile (PAN)–LiCF_3_SO_3_ nanocomposites containing dodecyl-amine-modified montmorillonite clay was investigated by Sharma et al. [17]. The nanocomposite showed an ion conductivity improvement by one order of magnitude at room temperature and 80 °C and the enhancement of cationic transport by polymer intercalation between the clay lamellae. The same author studied the ion–ion and polymer–ion interactions in such nanocomposites [18]. Modified clay aided in the dissociation of a salt, resulting in improved conductivity compared to the pristine PAN. Exfoliated nanocomposites were obtained from poly(4,4′-diphenylether-5,5′-bibenzimidazole) and organic-modified MMT [19]. The authors observed that the material showed improvement in thermal stability and mechanical properties. The protonic conductivity increased proportionally to the addition of clay. In 2012, Tang et al. [20] added PEO powder to an MMT solution containing carbon nanotubes and LiClO_4_. FTIR spectra showed that 95.7% of ClO_4_^−^ ions remained dissociated in the sample containing 10% of clay, which had the highest ionic conductivity. At the same time, pristine PEO had only 82.2% of dissociated anions. PVDF-HFP-natural Wyoming montmorillonite/smectite clay nanocomposite fibrous membrane was characterized, by AC impedance frequency analyzer between Li electrodes, to simulate its use in batteries [21]. Gaur et al. [22] produced proton-conducting polymer electrolyte nanocomposite using poly(vinyl alcohol) (PVA), chitosan, poly(styrene sulfonic acid) and MMT Cloisite^®^ 30B clay for application in methanol fuel cells. The properties of the membrane have been compared with Nafion^®^ 117. Zinc triflate doped poly(ε-caprolactone) incorporated with octa-decyl-amine-modified montmorillonite showed increased ionic conductivity and decreased activation energy with the addition of clay loading. The sample with the highest ionic conductivity showed better biodegradability over 90 days compared to the sample without loading polymer [5]. Cloisite^®^ organically modified with three distinct surfactants was also added to poly(4,4′-diphenylether-5,5′-bibenzimidazole) matrices to study the organic/inorganic interface of polymer nanocomposites. Singha and Jana [23] observed that the leaching of phosphoric acid, the proton source, decreased with the presence of clay and increased proton conductivity as its concentration increased. Plasticized nanocomposite SPEs were characterized by Choudhary et al. [24]. PEO-PMMA blend matrix was doped with LiCF_3_SO_3_, plasticized with propylene carbonate and mixed with MMT by solution cast and ultrasonic-microwave irradiation methods. The ionic conductivity of the samples showed Arrhenius-type behavior.

Gellan gum is a linear anionic exopolysaccharide consisting of α-1,4-L-rhamnose, β-1,3-D-glucose and β-1,4-D-glucuronic acid in a molar ratio of 1:2:1. Its native form has a high content of acyl, L-glyceryl and acetyl groups. On the other hand, its deacetylated form has a low content of acyl groups and this decreases its gelling and water absorption properties [25,26]. Gellan gum has been investigated for several applications including biomedical [27], medical and pharmaceutical [25] hydrogels for controlled fertilizer release [28] and soil humidity control [6], as well as solid polymeric electrolytes for electrochemical devices [29,30], among others.

Aiming to advance on NPE development, we studied the influence of the addition of different amounts of lyophilized MMT Na^+^SYN-1 on the ionic conductivity of NPEs made from gellan gum. We characterized the nanocomposites by electrochemical impedance spectroscopy, thermal analyses of DSC and TGA, UV-vis reflectance and transmittance and EPR spectroscopies.

## 2. Results and Discussion

Figure 1 shows the Nyquist diagrams of the imaginary impedance (Z″) versus real impedance (Z′) obtained from the electrochemical impedance spectroscopy measurements of GGLA-G40 and the NPEs GG-MMT10 and GG-MMT40. Increasing temperature leads to a decrease in the semicircle size, indicating a decrease in electrolyte resistance and an increase in conductivity [31]. The disappearance of semicircles in Nyquist diagrams at 55 °C for GGLA-G40, 40 °C for GG-MMT10 and 45 °C for GG-MMT40 indicates that there was a purely diffusional behavior above these temperatures [32]. Instead of a centered semicircle on the Z′ axis, a flat-shaped semicircle emerged. All diagrams containing a high-frequency semicircles have a point outside the curvature—around 1 MHz—except for the GGLA-G40 sample, in which disparity occurs at 158.489 kHz. This nonconformity can be associated with the formation of two semicircles, where the second is followed by a straight line, or because of electrical network interference in the measurement [33]. The first semicircle positioned on the left may refer to electrolyte bulk resistance and the surface impedance of the film, while the second may be related to ion transport. The straight line is due to the ion diffusion process [33].

Figure 2 presents the ionic conductivity values as a function of the inverse temperature for the samples of GGLA-based SPEs and NPEs.

From Figure 2 it is seen that all results exhibit linear behavior, which means that the charge transport mechanism is dominated by the hopping of the ions between the coordination sites, similar to behavior observed by others [34]. Likely, the ionic transport is assisted by the movement of polymer chains [35]. To fit these results using the Arrhenius model, the use of two straight lines was needed. The sample with the best ionic conductivity at low temperatures (<55 °C) was GGLA-G40. It exhibited values of 2.14 × 10^−6^ S/cm at 30 °C and 3.10 × 10^−4^ S/cm at 80 °C (Figure 2a). This was probably due to the high Li^+^ concentration that improved the ionic conductivity of the system. This usually occurs with solid polymeric electrolytes produced from polysaccharides with plasticizers and saline additives [26,36]. Gellan gum electrolytes without plasticizer reported by Noor et al. [26] presented 5.4 × 10^−4^ S/cm at room temperature. This value is higher when compared to the conductivity of the GGLA-G40 sample. Gellan gum membranes tend to form crystalline structures [30]. Thereby, the addition of glycerol to the electrolytes did not help the formation of a non-crystalline or semi-crystalline structure achieved in previous research [30]. On the other hand, formaldehyde is known as a crosslinking agent for polysaccharides [37]. Therefore, both glycerol and crosslinks may have hindered ion conduction by the material.

Having exhibited the best ionic conductivity, GGLA-G40 was selected for the addition of Na^+^SYN-1 clay as the basis for NPEs. Like GGLA-SPEs, Figure 2b shows that NPEs also followed the Arrhenius model of conductivity. GG-MMT1 presented the lowest ionic conductivity values of 1.14 × 10^−6^ S/cm at 30 °C and 2.11 × 10^−4^ S/cm at 80 °C. GG-MMT10 had the best ionic conductivity of 1.86 × 10^−5^ at 30 °C and GG-MMT40 had the best ionic conductivity of 3.99 × 10^−4^ at 80 °C. It is quite clear that GG-MMT5, GG-MMT40 and GG-MMT10 NPEs exhibit ionic conductivity higher than GGLA-G40, as expected and described by other authors [5,9] but lower when compared with 1.35 × 10^−4^ S/cm for PEO:KI/I_2_ dopped with 10 wt.% of TiO_2_ [38]. Although the ionic conductivity at room temperature of the GG-MMT10 was lower compared to those of Noor et al. [27], it was higher than the electrolytes with ionic liquid of Neto et al. [30]. The differences in the conductivity of the gellan gum-based samples may be due to different salt used and the presence of crosslinks. Ghadami et al. [7] studied the ionic conductivity of nanocomposites produced using gelatin reinforced with different amounts of distinct clays. Among these added clays the sample with 0.01 g Li^+^MMT showed the highest conductivity values of 8.25 × 10^−7^ S/cm at 22 °C and 2.09 × 10^−5^ at 90 °C and this was lower than that obtained by GG-MMT10. However, a common point between the two mentioned works is the fact that both presented a limit value for conductivity as a function of the amount of clay. After a certain concentration of clay, the mobility of the polymer chain can be hindered, leading to the suppression of ion transport.

Table 1 and Table 2 show the ionic conductivity from gellan gum electrolytes at 30 and 80 °C, respectively. The activation energy (Ea) of each sample is also exhibited in both kJ/mol and eV. Since two Arrhenius model fittings were applied to each sample, Table 1 shows the Ea of the electrolytes at low temperatures (from 30 to ~50 °C) and Table 2 contains Ea values for the highest temperatures (from ~50 to 80 °C). Ea went from 102.40 kJ/mol (GGLA-G10) to 191.85 kJ/mol (GGLA-G20) (Table 1) and from 50.81 kJ/mol (GGLA-G0) up to 68.12 kJ/mol (GGLA-G5) (Table 2). These values are analogous to natural macromolecules-based bio-blends reported by Mattos et al. [39]. Certainly, there was a strong interaction between macromolecules and Li^+^, restricting the movement of ions.

The increase in Ea with ionic conductivity for the samples GGLA-G0–GGLA-G5 (Table 2) may have been due to the formation of ionic aggregates, which would need more energy to cross the conductivity barrier. At this moment we must remember that besides Li^+^ there are also bigger Na^+^ from clay that may contribute to this conductivity. Meantime, there is a possibility that the changes in ionic conductivity occurred because of the restrained polymer chain mobility due to the presence of clay [1] and crosslinks. We can notice that there is a change in the Arrhenius fitting slopes at 45 and 60 °C, a temperature range that followed the disappearance of semicircles in Nyquist diagrams. This means that at these temperatures Li^+^ ions change their movement from charge transfer to diffusion, which would be associated in some way with the modification of the polymeric matrix structure, perhaps the fusion of crystalline regions or even chain reorganization [40].

Table 3 and Table 4 show the ionic conductivity of nanocomposites measured at 30 and 80 °C. Again, the results were fitted by the Arrhenius model and using two straight lines. Table 3 presents the results for low temperatures while Table 4 refers to higher temperatures. Ea of the NPEs samples decreased with increasing ionic conductivity of the samples, a similar trend seen by Halim et al. [34]. GG-MMT10 at the lowest temperatures and GG-MMT40 at the highest temperatures exhibited the highest conductivity values. Although gelatin-NPEs [7] had much lower activation energy values (~7.2 kJ/mol), they did not exceed the ionic conductivity of GGLA/Na^+^SYN-1 electrolytes. This makes it evident that the energy barrier of the present NPEs was not a hindrance to ion transport.

To verify the supposed structural reorganization of the polymer at temperatures around 50–60 °C, we measured the ionic conductivity as a function of temperature in the range of 30 to 80 °C at 10 °C intervals of the GG-MMT5 sample (Table 5). According to Zou et al. [2], the load added to a polymeric matrix prevents the crystalline portion of the latter from rearranging after heating with subsequent cooling. However, the gellan gum structure and conformation besides being sensitive to temperature, polymer concentration and aqueous environment is also very sensitive to the presence of mono- and divalent cations in the solution [41]. Figure 3 shows the ionic conductivity results as a function of an inverse of temperature for the GG-MMT5 sample. It is noted that the first measurement was fitted by two straight lines using the Arrhenius model. The intercept of these two lines was found at around 50 °C. At low temperatures, gellan forms an ordered helix of double strands, while at high temperatures it changes to a single-stranded polysaccharide and the viscosity of its solution decreases. This transition usually occurs between 35 and 50 °C [41]. The second measurement was fitted with only one straight line, where Ea was 46.29 kJ/mol (0.48 eV), close to that determined in the first series of measurements (Table 4) at temperatures above 50 °C. As the samples are loaded with Li^+^ and Na^+^ ions it probably promoted a change of gellan gum conformation.

Figure 4 shows thermal analyses of GGLA SPEs and NPEs. As seen in Figure 4a, the Na^+^-SYN-1 lyophilized clay suffered an initial mass loss of 5% from ~30 to 189 °C, presumably due to moisture loss. Two more events, the first up to 544 °C and the second up to 684 °C, with 5% and 4% mass decrease, respectively, were caused by dehydroxylation of the material. The total mass loss was 13%, leading to an ash content of 87%, surely consisting of silicates and aluminates [42]. Gellan gum-based samples presented very similar thermograms. For example, GGLA-G40 lost 9% of its mass from room temperature (~30 °C) to approximately 135 °C and it was certainly promoted by the loss of absorbed water. From 240 to 320 °C the main step of degradation of the solid polymer electrolyte occurred, with mass loss of ~45%. The derivative of these results, shown in Figure 4b, displays two peaks in this temperature range, one at 262 and the second at 300 °C (red line in Figure 4b). The hydrolysis and oxidation stage of this sample succeeded up to 562 °C with a 20% mass decrease. Finally, the pyrolysis of the polymer was verified, with a mass decrease of another 20%, resulting in an ash content of 6%. The NPEs samples presented water release with mass loss of 10% from room temperature (~30 °C) up to 130 °C, except for GG-MMT20, which suffered a mass decrease up to 174 °C. Mass losses of 47% for GG-MMT20 and 53% for GG-MMT15 were observed up to 370 °C. This occurred possibly because of hydrolysis and oxidation of the gellan gum. A second degradation stage lasting up to 600 °C caused a 10% mass loss, perhaps promoted by polymer pyrolysis. A third stage that went to 706 °C for GG-MMT20 and 883 °C for GG-MMT5 generated a 10% mass loss. Supposedly this event was caused by clay dehydroxylation [43]. After a constant mass loss, the samples exhibited ash content of 2.8% for GG-MMT1 and 25% for GG-MMT40, indicating an increase with the increase of added clay. Gellan gum/Na^+^SYN-1 NPEs did not exhibit any improvement in the thermal stability of SPEs, as the deterioration temperatures of the SPEs or NPEs, remained practically within the same range, i.e., between 124 and 174 °C for GG-MMT15 and GG-MMT20, respectively.

The DSC measurements of the samples GGLA-G40, GG-MMT10 and GG-MMT40 did not show any thermal event in the temperature range from −60 to 80 °C (Appendix A). 

The XRD diffractogram of Na^+^SYN-1 clay shows that it has a basal spacing of approximately 11.2 Å (2θ ≈ 7.96°), which is in agreement with the literature (Figure 5). GG-MMT1 to GG-MMT20 nanocomposites presented intercalated structure, with the basal spacing of 11.4 Å (2θ ≈ 7.78°) for GG-MMT10 and 12.8 Å (2θ ≈ 6.90°) for GG-MMT1 and GG-MMT20. The presence of gellan gum did not promote the exfoliation of clay layers, unlike observed by others [44]. While GG-MMT1 shows a peak at 2θ = 6.90°, the other nanocomposites showed two peaks: one at 2θ ≈ 7.78° related to the clay basal spacing and the other at 2θ = 6.90° related to the spacing of the lamellae displaced by the polymer. Considering that the 2θ = 7.78° peak, absent in GG-MMT1, heightens with increasing clay concentration, it is assumed that the GG-MMT1 electrolyte is the unique predominantly intercalated nanocomposite [45].

A noteworthy point is a polymer-clay interaction. If the polymer macromolecules have a negative charge they are not adsorbed onto the surface of the clay material, which has negative charge [27]. Thereby, polymer chains are unlikely to separate the layers, leading the nanocomposite to have an exfoliated structure. Considering that both gellan gum and Na^+^SYN-1 have negative net charges, the interaction between them would certainly result in a mostly intercalated or intercalated-flocculated structure, corroborating that observed in Figure 5 diffractograms. Finally, scanning electron microscopy (SEM) images (not shown here) also revealed a smoother surface of the sample GG-MMT40 when compared to GGLA-G40, due probably to the decrease of its crystallinity after the addition of clay.

Another important characteristic of polymer electrolytes is their transparency and/or opacity. Therefore, NPEs were subjected to UV-vis transmittance and reflectance analyses and the results are shown in Figure 6.

As seen in Figure 6a, GGLA-G40 is the most transparent among all analyzed samples in the whole spectrum range and its transparency increased from 44% at 400 nm to 58% at 800 nm. As expected, the NPEs are opaquer and this increases with Na^+^SYN-1 concentration [35]. As observed by Zeng et al. [3] and Ray and Okamoto [46], there should be no decrease in transmittance with an increase in clay concentration in a polymer matrix. Meanwhile, as seen in Figure 6a, the addition of Na^+^SYN-1 promoted a gradual decrease in the transmittance of the samples. This can be due to the larger size of clay nanoparticles that can obstruct the passing light. The XRD diffractograms (Figure 5) evidenced that part of the clay remained flocculated in the electrolyte and the concentration of flocculated particles increased with the increased clay loading. Thus, we can assume that these clusters may be responsible for the lower transmittance values of the NPEs. Overall, these transmittance values are low compared to alginate-based NPEs [35] but similar to gelatin-based NPEs [1]. Reduced transmittance of gellan gum-based materials may also be related to the association and order of polymer chains. GG-MMT10, which had the best conductivity at room temperature, had at most 42% of transmittance at 800 nm. GG-MMT40 had the lowest transmittance value of 22% at 800 nm.

As predicted, the percentage of optical reflectance tends to increase as a function of clay concentration, contrary to that observed in transmittance (Figure 6b). The highest reflectance percentages were seen between 200 and 400 nm, showing that the membranes can reflect approximately 11% (GG-MMT1 at 400 nm) to 25% (GG-MMT40 at 200 nm) in the UV band. GGLA-G40 showed three reflectance peaks: at 203 and 227 nm—both with 19%—and at 311 nm with 15%. GG-MMT1 exhibited the lowest reflectance peaks, with 20% at 225 nm and 14% at 311 nm. On the other hand, GG-MMT40 exhibited 24% reflectance at 226 nm and approximately 19% at 316 nm. In the visible range, the samples showed a decrease in reflectance, displaying values from 8% (GG-MMT5) to 12% (GG-MMT15) at 800 nm. GG-MMT10 demonstrates UV reflectance of 22% at 228 nm, 17% at 313 nm and a gradual decrease from 13% at 401 nm to 10% at 800 nm. The peaks observed around 200 and 226 nm are related to the σ–σ* transition of the tetrahedral sites of the sp^3^-hybridized carbons belonging to the polymer. However, the peak at 317 nm refers to the π-π* transition of the C=O sp^2^ carbons [47]. Another factor that may be linked to decreased transmittance and increase in reflectance due to clay loading is the thickness of the films [48]. Table 6 lists the GGLA-G40 and NPEs thicknesses, measured in triplicates with a Mitutoyo Micrometer thickness gauge. GG-MM10 with 0.085 mm and GG-MMT15 with 0.080 mm exhibited the highest thickness values compared to the other electrolytes. These presented the highest reflectance percentages, just below GG-MMT40 and above GG-MMT20.

The coordination geometry of the Cu^2+^ ions in the GGLA NPEs was determined by Continuous-Wave Electron Paramagnetic Resonance spectroscopy (CW-EPR). Figure 7 shows the X-band CW-EPR spectra measured at room temperature of GGLA-base nanocomposite for four different copper concentrations. As can be seen in Figure 7, the shape of the EPR spectra strongly depends on the Cu(ClO_4_)_2_ content of the samples. The samples doped with the lowest Cu(ClO_4_)_2_ concentration exhibit an anisotropic EPR spectrum characteristic of paramagnetic Cu^2+^ ions located in axially distorted sites, superimposed to a broad background. The Cu^2+^ ion has a 3d^9^ electronic configuration and spin S = 1/2. The nuclear spin, for both ^63^Cu (natural abundance 69%) and ^65^Cu (natural abundance 31%) isotopes, is *I* = 3/2. The dipole–dipole interaction between the magnetic moment of the nucleus and the electronic moment of the paramagnetic ion split each resonance line into (2*I* + 1) = 4 hyperfine components. Therefore, it is expected to observe four hyperfine satellites in the low field part of the Cu^2+^ EPR spectra and four hyperfine satellites in the high field part of the spectra. A set of four evenly spaced hyperfine lines is clearly observed in the low field part of the spectra (around 2800 G) for the samples with a lower copper concentration in Figure 7. Nevertheless, in the high field part of the spectra (≈3300 G), the hyperfine satellites are not resolved and a single line is observed. For the samples doped with higher copper concentration, the hyperfine structure and the anisotropy related to the g-tensor are smeared out leading to a single isotropic line that resembles a Lorentzian line shape.

The spectra obtained for GGLA-MMT10 and GGLA-MMT40 samples are similar to those obtained in Figure 7 and are shown in the Appendix A. A closer examination of the EPR spectra in Figure 7 reveal that the experimental spectrum is a superposition of two components, an axial structured spectra due to isolated Cu^2+^ ions and a structure-less anisotropic signal due to the presence of magnetically coupled spins. Considering the hyperfine interaction, the isolated spins EPR spectrum can be described in terms of the spin-Hamiltonian for d^9^ ions in axial symmetry:(1)$$H=g_{∥}\beta H_{z}S_{z}+g_{⊥}\beta \left(H_{x}S_{x}+H_{y}S_{y}\right)+A_{∥}I_{z}S_{z}+A_{⊥}\left(I_{z}S_{z}+I_{y}S_{y}\right)$$

In this equation, *z* is the tetragonal symmetry axis, *β* is the Bohr magneton; *H* is the applied magnetic field; *S* and *I* are the electron and nuclear spin operators. The *g* and *A* tensors have the components *g*_∥_ = *g*_z_, *g*_⊥_ = *g*_x_ = *g*_y_ and *A*_∥_ = *A*_z_, *A*_⊥_ = *A*_x_ = *A*_y_, respectively. The first two terms in Equation (1) represent the Zeeman interaction, i.e., the interaction between the electronic spin and the magnetic field. The last two terms represented the hyperfine interaction, i.e., the coupling between the electronic and nuclear spins. In this work, parameters *A*_∥_ and *A*_⊥_ are given in units of frequency (or energy). It is assumed that both tensors are diagonal in the same coordinated system.

The experimental Cu_2_^+^ EPR spectra were analyzed by numerical simulation of the spin Hamiltonian of Equation (1) by using the Easy-Spin package [49]. Fitting parameters for the anisotropic spectra include *g*_∥_, *g*_⊥_, *A*_∥_ and *A*_⊥_. Fitting parameters describing broadening are also included. Isotropic broadening, named *lwpp* (in units of magnetic field), describes the convolution of the spectrum with a Gaussian and/or Lorentzian line shape. Peak-to-peak (*pp*) refers to the magnetic field difference between the maximum and minimum of a first derivative line shape. Another necessary fitting parameter is the anisotropic broadening named HStrain, or simply *HS,* expressed in units of frequency (or energy), describing broadening due to unresolved hyperfine couplings or other transition-independent effects [49].

Figure 8 shows the experimental and the simulated spectra of the less concentrated samples. The agreement between both spectra is satisfactory since the position and the intensities of the prominent features are well reproduced. The Hamiltonian parameters, deduced from the simulated spectra are collected in Table 7. To fit the experimental spectra, two components were necessary, represented in blue (isolated spins) and in green (coupled spins). The sum of both is in red, lying over the experimental data (black) almost exactly.

The fact that the component associated to coupled spins is anisotropic indicates that the magnetic coupling has a predominantly dipolar origin. The dipolar coupling is strong enough to broaden the hyperfine structure but not enough effective to smear out the g-anisotropy. When the copper concentration increases, the average distance between copper ions decreases, favoring the exchange effect and leading to an isotropic spectrum, as can be seen in Figure 7.

The Hamiltonian parameters in Table 7 give interesting information regarding the ligand type and the coordination geometry of the copper ion in the membranes. In the present investigation, it is observed that as *g*_∥_ > *g*_⊥_ > *g_e_*, where *g_e_* = 2.0023 is the free electron *g*-value. Therefore, it can be concluded that the Cu^2+^ ions are located in tetragonally distorted octahedral sites, the d_x2-y2_ orbital (B_1g_ state) being the ground state.

The spectroscopic ratio *f* = *g*_∥_/*A*_∥_ is often used as empirical evidence of tetrahedral distortion around the Cu^2+^ ion in copper complexes. For planar complexes, *f* is in the range 110–120 cm and for moderate tetrahedral distortion the range is 130–150 cm. Higher *f* values indicate considerable distortion [50,51]. The *f* value of the investigated GGLA NPEs is *f* = 177 cm. The value derived for the PCN formed by plasticized gelatin and montmorillonite clay doped with Cu^2+^ is around *f* = 154 cm for the first Cu^2+^ component and *f* = 136 cm for the second one [1]. The high *f*-value of our nanocomposites suggests accentuated tetrahedral distortion in the copper coordination sphere.

The EPR parameters also provide information on the nature of metal-ligand bonds in copper complexes. The bonding coefficient *α*^2^ (i.e., the in-plane σ bonding) can be evaluated from the EPR spin Hamiltonian parameters by using the simplified expression in Equation (2) [52,53].
(2)$$\alpha^{2}=\frac{A_{∥}}{P}+\left(g_{∥}-2.002\right)+\frac{3}{7}\left(g_{⊥}-2.002\right)+0.04,$$
where *P* = 0.036 cm^−1^ is the dipolar hyperfine coupling constant for free Cu^2+^. The values of this coefficient range from a minimum value of 0.5 for a completely covalent copper−ligand bond up to a maximum value of 1.0 for a completely ionic bond [54,55]. The value of *α*^2^ for our GGLA NPEs, *α*^2^ = 0.79 (Table 8), is comparable to those derived for the Gel—Sca-3Na NPEs, *α*^2^ ≈ 0.85 [1] suggesting a moderate covalence for the σ-bonding of the copper ion in these polymer-clay nanocomposites.

## 3. Materials and Methods

During this research, we used synthetic mica-montmorillonite SYN-1 (Barasym SSM-100, NL Industries), HCl (Synth, P.A. ACS, M.M. 36.46, purity 36.5%), NaOH (Neon, M.M. 40.00 g/mol; purity 97.7%), NaCl (Synth, P.A. ACS; M.M. 58.44 g/mol, purity 99%), AgNO_3_ (Tec-Lab, P.A.), low acyl gellan gum (CP Kelco, KELCOGEL CGLA), LiClO_4_ (Êxodo Científica, P.A. ACS, M.M. 106.39 g/mol, purity 95%), formaldehyde (Vetec, P.A. ACS, M.M. 30.03 g/mol, purity 36.5–38%) and glycerol (Vetec, P.A. ACS, M.M. 92.1 g/mol, purity 99.5%). 

Gellan gum electrolytes were prepared as described elsewhere [26]. 1 g of gellan gum (GG) was placed in each of the seven 250 mL beakers. Then, 150 mL of Millipore Milli-Q^®^ water with controlled resistivity of 18 mΩ^−1^ cm^−1^ at 25 °C was added to each beaker and the resulting mixtures were magnetically stirred for 1 h at room temperature to complete homogenization. Next, 0.125 mL formaldehyde, 0.40 g glycerol and 0.01–0.04 g of LiClO_4_, were added to the GG dispersions and continued under magnetic stirring for the next 24 h. After that, the samples were transferred to acrylic Petri dishes and placed in a Fanem model 520 stove at 40 °C for drying until transparent membranes were formed. The membranes were stored in a desiccator for future analyses. The composition of the samples is displayed in Table 9.

For purification and cationic exchange of NH_4_^+^ to Na^+^ in the clay. 40 g of SYN-1 was added in 4 L of deionized water and this mixture was stirred for 2 h. 2 mol/L HCl was added under stirring until the pH reached 3.5 for the removal of carbonate ions (CO_3_^2−^). The dispersion was stirred for approximately 20 min and centrifuged in a Hitachi model CR22GIII centrifuge at 10,000 rpm at 25 °C for 30 min to remove water-soluble salts. The supernatant was discarded and the pellets were suspended in 4 L of distilled water. NaOH solution was added to the stirring dispersion until reaching pH 8. The dispersion rested for 12 h. The supernatant was siphoned and its pH was adjusted to 3.5 using 2 mol/L HCl. 160 g of NaCl was added to the supernatant for flocculation. The pellets were again suspended in deionized water and the pH was adjusted to 8 using NaOH. The dispersion with pH 3.5 was allowed to flocculate for 12 h. This procedure was repeated until the clay dispersion, having pH 8, was translucent. Thus, the pH 8 dispersion and the supernatant of the pH 3.5 clay solution were discarded. The remaining clay was centrifuged at 10,000 rpm and 25 °C for 30 min. Then, it was placed for dialysis in deionized water for purification until a negative test for chloride ions. This test was performed by adding 0.1 mol/L AgNO_3_ to an aliquot of the suspension. Finally, the purified clay was lyophilized using Thermo Electron Corporation Model Moduly0D equipment and, finally, stored.

Membranes were prepared in triplicate by solution method. Initially, 20 mL of Millipore Milli-Q^®^ water with controlled resistivity of 18 mΩ^−1^ cm^−1^ at 25 °C was added to six 50 mL beakers, followed by the addition of different amounts of SYN-1 in each. The mixture was stirred for 24 h. Meanwhile, a polymer matrix suspension was prepared based on the best ionic conductivity result. 6 g of GGLA was weighed into a 1000 mL beaker, 780 mL Millipore Milli-Q^®^ water was added and the resulting mixture was stirred for 24 h at room temperature (~25 °C) for complete homogenization. Subsequently, 2.4 g of LiClO_4_, 0.75 mL of formaldehyde and 2.4 g of glycerol were added to the GGLA suspension under stirring. The polymer suspension was kept under vigorous magnetic stirring for 2 h for homogenization. The GGLA suspension was divided into six beakers. The previously prepared clay suspensions were added to the polymeric suspension. Mixtures of gellan gum and Na^+^SYN-1 were kept under magnetic stirring for 24 h, transferred to acrylic Petri plates and placed in a stove at 40 °C for drying. Finally, the membranes were stored in a desiccator for future analyses. Gellan gum/Na^+^SYN-1 clay membrane compositions are shown in Table 10. Figure 9 shows the GG-MMT10 nanocomposite sample.

The electrochemical impedance spectroscopy technique was used to determine the ionic conductivity (σ) of the NPE samples. The ideal, non-capacitive resistance was read from the semicircle intercept with the real impedance axis (Z′) of the Nyquist diagram and used to obtain the ionic conductivity values from Equation (3).
(3)$$\sigma =\frac{l}{RA}$$
where *l* is the membrane thickness (cm), *A* is the membrane area (cm^2^) and *R* is the resistance (Ω) obtained from the impedance measurements. The electrochemical impedance spectroscopy measurements were performed as a function of the temperature increase. 

For these measurements, the NPE membranes had thicknesses ranging between 0.055 and 0.085 mm. They were cut in 1.54 cm^2^ circles, pressed between two polished stainless-steel electrodes and placed in a Teflon^®^ cell coupled to a vacuum pump. The electrical contacts were the metal bottom of the cell and a stainless-steel tube. A thermocouple was placed inside the tube and the temperature was read in the range of 30 (303 K) to 80 °C (353 K) at 5 °C intervals. The samples were packed at low pressure to allow good contact and avoid humidity influence. The Teflon^®^ cell was placed in the bottom of an EDG 1800 5P muffler with temperature and time control. Impedance measurements were made using a Solartron model 1260 impedance meter in the frequency range of 0.1 Hz to 10 MHz with 5 mV amplitude.

Thermogravimetry (TGA) measurements were made to evaluate the thermal stability of Na^+^SYN-1 clay, GGLA SPEs and NPEs. These analyses were done from room temperature to 1000 °C (273 K) at a heating rate of 15 °C/min and under nitrogen (N_2_) atmosphere on TGA-Q50 equipment from TA Instruments.

Differential Scanning Calorimetry (DSC) was used to determine the thermal transitions of GGLA/Na^+^SYN-1 nanocomposites. The procedure was performed on a DSC Q2000 from TA Instruments at 50 mL/min N_2_ flow. Three temperature ramps were applied. The first ramp was programmed from ~40 to 100 °C at a heating rate of 20 °C/min to eliminate possible solvents in the samples. The second from ~40 to −60 °C and the third from −60 to 80 °C at a heating rate of 15 °C/min to check for the presence or absence of Tg.

UV-vis spectroscopy measurements were performed on the samples on a Jasco V-670 equipment within the wavelength range of 200 to 800 nm.

The electrochemical impedance spectroscopy measurements, as well as the thickness measurements of the pure and nanocomposite electrolytes, were performed in triplicates. The standard deviation calculations were made using the Microsoft^®^ Excel program.

The samples were characterized by XRD in a Rigaku Ultima IV equipment at a voltage of 40 kV, 40 mA current, λ(CuKα) = 1.540 Å, 2–80° (2*θ*) angle range, step 0.0200/s, 1°/min over a polished silicon wafer. Through X-ray diffraction, it is possible to find the distance between the intercalated lamellae (basal spacing) of Na^+^SYN-1 clay or nanocomposites and to detect exfoliated structures of nanocomposites using Bragg’s law Equation (4).
(4)λ=2d senθ,
where *λ* is the X-ray wavelength, *d* is the basal spacing and *θ* is the angle.

Continuous-wave (CW)-EPR spectra were obtained for the NPE samples containing Cu(ClO_4_)_2_ at different concentrations. These measurements were done at room temperature with a Bruker Elexsys E580 spectrometer operating at an X-band frequency (≈9.43 GHz). The modulation frequency is 100 kHz. To avoid line shape distortions, modulation amplitude and lock-in time constant were set to optimum values. Microwave power was selected in the linear region where no line shape saturation is observed.

## 4. Conclusions

The present work presented the results of the preparation and characterization of GGLA SPEs and NPEs containing Na^+^SYN-1 clay. Electrochemical impedance spectroscopy revealed that the electrolyte with the best ionic conductivity was GG-MMT10, which reached 1.86 × 10^−5^ and 3.74 × 10^−4^ S/cm at 30 and 80 °C, respectively. Two adjustment lines by the Arrhenius model were applied to fit the results and they provided activation energies values of 99.50 kJ/mol (1.03 eV) in the temperature range from 30 to 45 °C and 29.20 kJ/mol (0.30 eV) from 45 to 80 °C. Another sample that can be highlighted is the GG-MMT40, which exhibited conductivity values of 1.38 × 10^−5^ and 3.99 × 10^−4^ S/cm at 30 and 80 °C, respectively. These results were also fitted using the two lines Arrhenius model. Thermogravimetry results revealed that the electrolytes have undergone almost no change concerning their stability with the incorporation of Na^+^SYN-1 clay. XRD indicated that the GG-MMT1 nanocomposite formed a predominantly intercalated structure, while the other electrolytes remained intercalated-flocculated. Transmission spectroscopy showed that the addition of Na^+^SYN-1 clay to the matrices of gellan gum decreased their transmittances in the range of 200 to 800 nm. The GG-MMT10 membrane exhibited a maximum transmittance of 42% at 800 nm, which dropped to 22% in the GG-MMT40 sample. The UV-vis reflectance spectroscopy showed that the clay concentration increased the reflectance of the electrolytes, having 8% for GG-MMT5 NPEs and 12% for GG-MMT15, both measured at 800 nm. EPR spectroscopy indicates that the copper distribution in the samples is not homogeneous, leading to the simultaneous presence of magnetically isolated copper ions and dipolar/exchange coupled ones. As the copper concentration increases, coupled ions dominate the EPR spectra. The inclusion of Na^+^SYN-1 clay does not change the shape of the EPR spectra, indicating no change in the local symmetry of copper ions. In summary, the results have shown that GGLA NPEs can be the target of studies in applications such as batteries and solar panels.

## Figures and Tables

**Figure 1 molecules-27-08721-f001:**
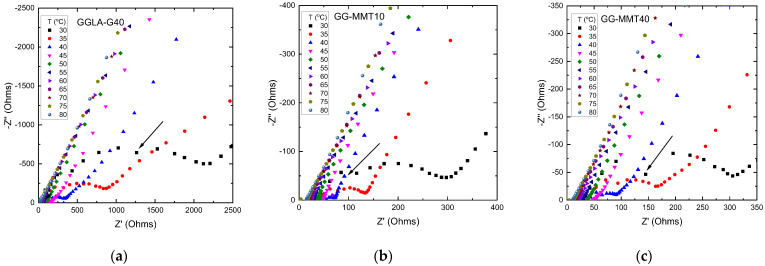
Nyquist diagrams of GGLA-G40 (**a**), GG-MMT10 (**b**) and GG-MMT40 (**c**) samples at temperatures ranging from 30 to 80 °C. Inset arrows indicate points outside the semicircle curvature.

**Figure 2 molecules-27-08721-f002:**
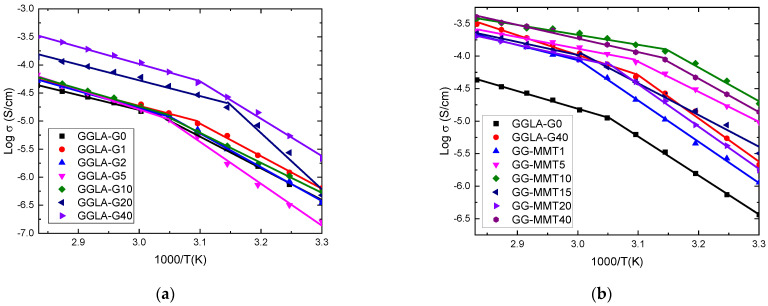
Log of ionic conductivity as a function of the inverse temperature of GGLA-based SPEs (**a**) and GGLA-G0, GGLA-G40 and GG-MMT-based NPEs (**b**) with different amounts of LiClO_4_.

**Figure 3 molecules-27-08721-f003:**
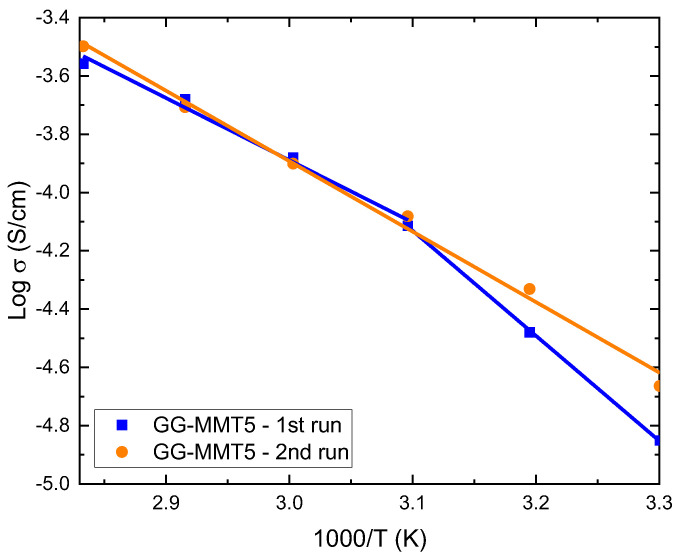
Log of ionic conductivity as a function of the inverse temperature of GG-MMT5 NPEs (solid symbols). The results were fitted with the Arrhenius model (solid lines).

**Figure 4 molecules-27-08721-f004:**
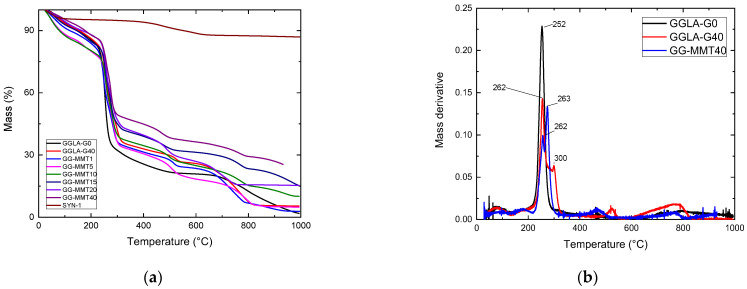
TGA of clay, GGLA-G0, GGLA-G40 and GGLA-MMT NPEs (**a**) and mass derivative of GGLA-G0, GGLA-G40 and GG-MMT40 (**b**).

**Figure 5 molecules-27-08721-f005:**
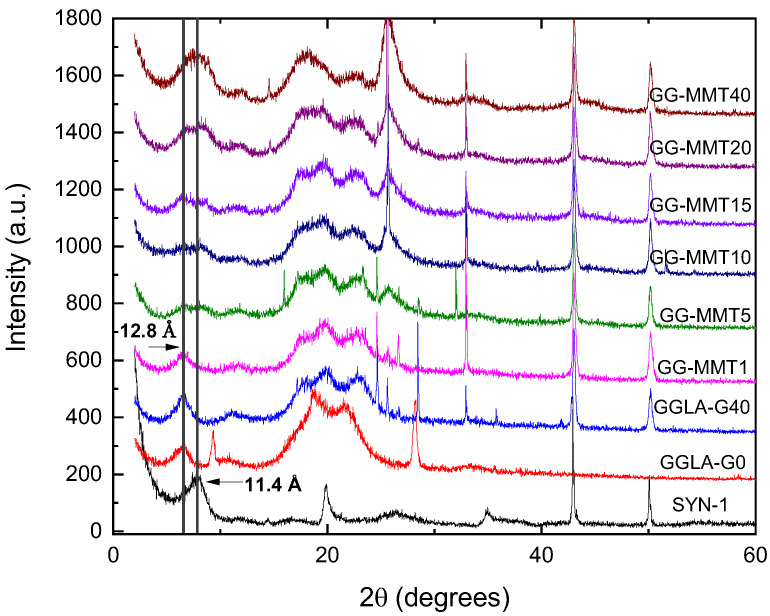
DRX diffractograms of Na^+^SYN-1 clay; GGLA-G0 and GGLA-G40 SPEs; and GG-MMT1 to GG-MMT40 NPEs.

**Figure 6 molecules-27-08721-f006:**
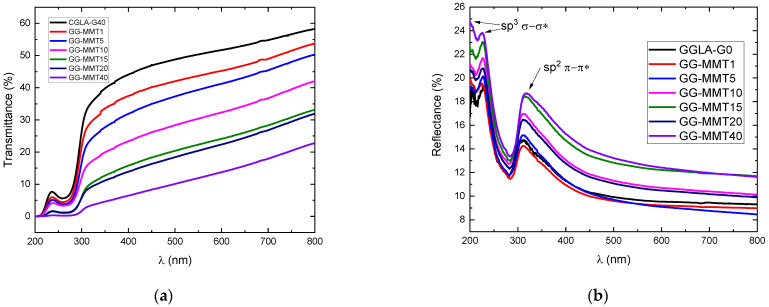
Optical transmittance (**a**) and reflectance (**b**) as a function of wavelength of GGLA SPEs and NPEs.

**Figure 7 molecules-27-08721-f007:**
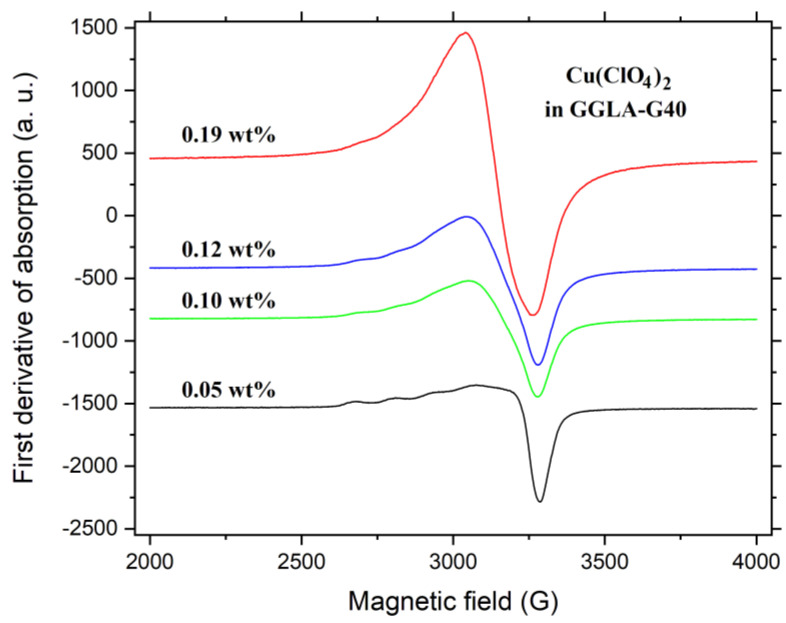
X-band CW-EPR spectra, measured at ≈300 K, of GGLA-G40 samples at different Cu(ClO_4_)_2_ concentrations. All spectra were measured at the same experimental conditions and normalized to the sample mass. The microwave frequency was ≈9.43 GHz.

**Figure 8 molecules-27-08721-f008:**
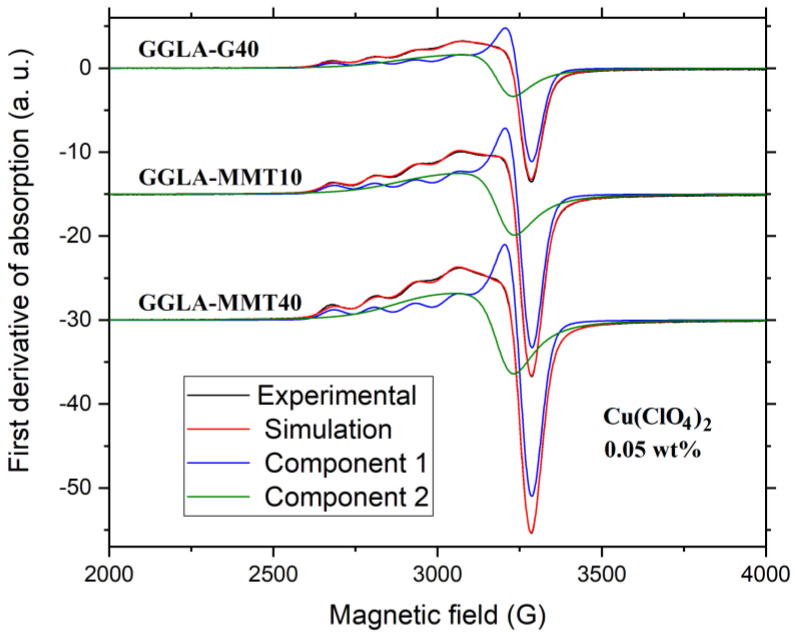
Experimental and simulated EPR spectra of the samples with 0.05 wt.% of Cu(ClO_4_)_2_. All spectra were measured at the same experimental conditions and normalized to the sample mass. The microwave frequency was ≈9.43 GHz.

**Figure 9 molecules-27-08721-f009:**
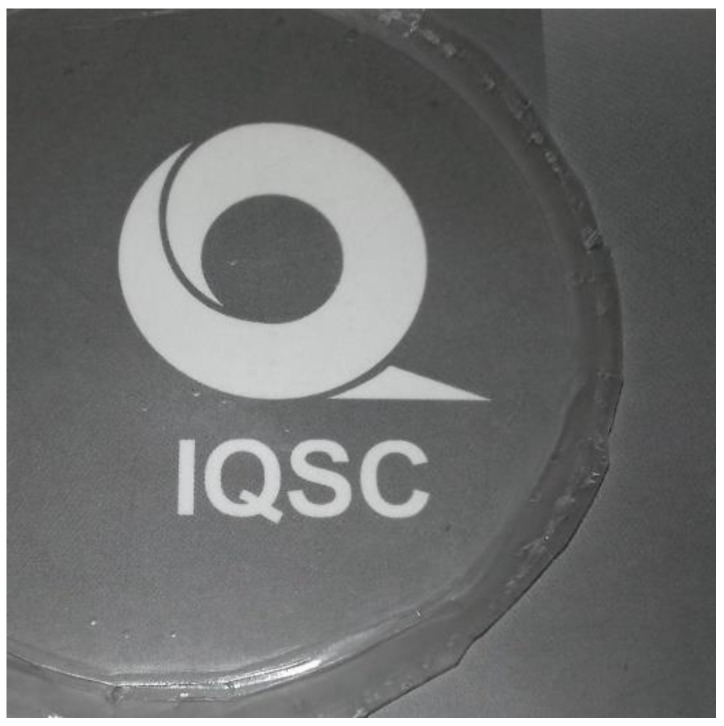
Picture of GG-MMT10 NPE membrane.

**Table 1 molecules-27-08721-t001:** Ionic conductivity and activation energy at low temperatures of GGLA SPEs.

Sample	σ (S/cm) at 30 °C	Temperature (°C)	Ea	
			(kJ/mol)	(eV)
GGLA-G0	3.64 × 10^−7^ ± 2.22 × 10^−7^	30–55	107.89	1.12
GGLA-G1	6.39 × 10^−7^ ± 4.47 × 10^−7^	30–50	110.91	1.15
GGLA-G2	3.48 × 10^−7^ ± 2.45 × 10^−7^	30–55	115.19	1.19
GGLA-G5	1.61 × 10^−7^± 0.99 × 10^−7^	30–55	142.88	1.48
GGLA-G10	5.50 × 10^−7^± 3.82 × 10^−7^	30–55	102.40	1.06
GGLA-G20	4.76 × 10^−7^ ± 4.60 × 10^−7^	30–45	191.85	1.99
GGLA-G40	2.14 × 10^−6^ ± 1.54 × 10^−6^	30–50	127.40	1.32

**Table 2 molecules-27-08721-t002:** Ionic conductivity and activation energy at high temperatures of GGLA SPEs.

Sample	σ (S/cm) at 80 °C	Temperature (°C)	Ea	
			(kJ/mol)	(eV)
GGLA-G0	4.32 × 10^−5^ ± 2.64 × 10^−5^	55–80	50.81	0.53
GGLA-G1	5.17 × 10^−5^± 3.66 × 10^−5^	50–80	54.43	0.56
GGLA-G2	5.20 × 10^−5^ ± 3.94 × 10^−5^	55–80	58.46	0.60
GGLA-G5	6.66 × 10^−5^± 1.06 × 10^−5^	55–80	68.12	0.71
GGLA-G10	5.87 × 10^−5^ ± 0.98 × 10^−5^	55–80	60.12	0.62
GGLA-G20	1.40 × 10^−4^ ± 1.13 × 10^−4^	45–80	53.64	0.56
GGLA-G40	3.10 × 10^−4^ ± 0.65 × 10^−4^	50–80	58.35	0.60

**Table 3 molecules-27-08721-t003:** Ionic conductivity and activation energy at 30 ^o^C of GGLA NPEs.

Sample	σ (S/cm) at 30 °C	Temperature (°C)	Ea	
			(kJ/mol)	(eV)
GG-MMT1	1.14 × 10^−6^ ± 0.61 × 10^−6^	30–60	121.46	1.26
GG-MMT5	9.57 × 10^−6^ ± 4.19 × 10^−6^	30–50	88.47	0.92
GG-MMT10	1.86 × 10^−5^ ± 0.30 × 10^−5^	30–45	99.50	1.03
GG-MMT15	3.19 × 10^−6^ ± 1.95 × 10^−6^	30–60	93.32	0.97
GG-MMT20	1.76 × 10^−6^ ± 1.42 × 10^−6^	30–55	124.77	1.29
GG-MMT40	1.38 × 10^−5^ ± 0.42 × 10^−5^	30–45	98.76	1.02

**Table 4 molecules-27-08721-t004:** Ionic conductivity and activation energy at 80 °C of GGLA NPEs.

Sample	σ (S/cm) at 80 °C	Temperature (°C)	Ea	
			(kJ/mol)	(eV)
GG-MMT1	2.11 × 10^−4^ ± 0.39 × 10^−4^	60–80	44.56	0.46
GG-MMT5	2.46 × 10^−4^ ± 0.10 × 10^−4^	50–80	34.08	0.35
GG-MMT10	3.74 × 10^−4^ ± 1.26 × 10^−4^	45–80	29.20	0.30
GG-MMT15	2.24 × 10^−4^ ± 1.27 × 10^−4^	60–80	38.55	0.40
GG-MMT20	2.03 × 10^−4^ ± 1.38 × 10^−4^	55–80	39.09	0.40
GG-MMT40	3.99 × 10^−4^ ± 0.83 × 10^−4^	45–80	40.11	0.42

**Table 5 molecules-27-08721-t005:** The activation energy of two consecutive ion conductivity measurements of the GG-MMT5 nanocomposite.

Sample	Temperature (°C)	Ea	
		(kJ/mol)	(eV)
GG-MMT5	30–50 (1st run)	69.04	0.71
	50–80 (1st run)	40.89	0.42
	30–80 (2nd run)	46.29	0.48

**Table 6 molecules-27-08721-t006:** Thicknesses of GGLA-G40 and GGLA/Na^+^SYN-1 NPEs.

Sample	Thickness (mm)
GGLA-G40	0.070 ± 0.009
GG-MMT1	0.070 ± 0.005
GG-MMT5	0.070 ± 0.005
GG-MMT10	0.085 ± 0.008
GG-MMT15	0.080 ± 0.003
GG-MMT20	0.065 ± 0003
GG-MMT40	0.065 ± 0.005

**Table 7 molecules-27-08721-t007:** Spin Hamiltonian parameters of the samples with 0.05 wt.% of Cu(ClO_4_)_2_, obtained from the simulation of the experimental EPR spectra. The values of *A* and strains *HS* are given in MHz. Gaussian (G) or Lorentzian (L) isotropic line width *lwpp* are given in mT. (The accuracy of determination of EPR parameters was estimated to be ±0.005 for g values and ±6 MHz for hyperfine splitting constants.

	GGLA-G40	GGLA-MMT10	GGLA-MMT40
Isolated spins			
*g* _⊥_	2.0716	2.0702	2.0704
*g* _∥_	2.3529	2.3492	2.3498
*A* _⊥_	12.3	11.7	11.9
*A* _∥_	399	396	396
*HS* _⊥_	180	190	191
*HS* _∥_	235	230	221
*Lwpp—G*	2.11	1.79	1.76
*lwpp—L*	1.24	1.25	1.28
*Weight*	1.0	1.0	1.0
Coupled spins			
*g* _⊥_	2.1124	2.1094	2.1094
*g* _∥_	2.2867	2.2796	2.2867
*HS* _⊥_	0.90	1.22	1.22
*HS* _∥_	771	743	674
*Lwpp—G*	0.15	0.57	050
*lwpp—L*	9.2	9.87	10.0
*Weight*	0.93	0.88	1.02

**Table 8 molecules-27-08721-t008:** The *g*_‖_ factors and *A*_‖_ values (in 10^−4^ cm^−1^), the parameter *f* of the tetrahedral distortion and the covalency parameter *α*^2^ for the studied 0.05 wt.% GGLA-G40 sample.

Samples	*g* _∥_	*A*_‖_ (10^−4^ cm^−1^)	*f* (cm)	*α* ^2^	References
0.05 wt.% GGLA-G40	2.353	133	177	0.79	This paper
Gel—SCa-3-Na (1)	2.360	153	154	0.85	[1]
Gel—SCa-3-Na (2)	2.308	170	136	0.85	[1]

**Table 9 molecules-27-08721-t009:** Compositions of GGLA-based SPEs.

Sample	Gellan Gum (g)	Glycerol (g)	LiClO_4_ (g)	HCHO (mL)
GGLA-G0	1.0000	0.4000	0.0000	0.125
GGLA-G1	1.0000	0.4000	0.0100	0.125
GGLA-G2	1.0000	0.4000	0.0200	0.125
GGLA-G5	1.0000	0.4000	0.0500	0.125
GGLA-G10	1.0000	0.4000	0.1000	0.125
GGLA-G20	1.0000	0.4000	0.2000	0.125
GGLA-G40	1.0000	0.4000	0.4000	0.125

**Table 10 molecules-27-08721-t010:** Compositions of GGLA-MMT-based NPEs.

Sample	Gellan (g)	Glycerol (g)	LiClO_4_ (g)	HCHO (mL)	Na^+^SYN-1 (g)
GG-MMT1	1.0000	0.4000	0.0100	0.125	0.0100
GG-MMT5	1.0000	0.4000	0.0200	0.125	0.0500
GG-MMT10	1.0000	0.4000	0.0500	0.125	0.1000
GG-MMT15	1.0000	0.4000	0.1000	0.125	0.1500
GG-MMT20	1.0000	0.4000	0.2000	0.125	0.2000
GG-MMT40	1.0000	0.4000	0.4000	0.125	0.4000
GG-MMT1	1.0000	0.4000	0.0100	0.125	0.0100

## Data Availability

The data presented in this study are available on request from the corresponding author.

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
