# Peer review of "Improved Conductivity in Gellan Gum and Montmorillonite Nanocomposites Electrolytes"

_molecules, 2022, doi:10.3390/molecules27248721_

Round 1

Reviewer 1 Report

This paper is complete in the sense that it contains no systematic study i.e. synthesis, characterization and  correlation. 

However I would like to suggest few modifications.     

1. In composition most of tables takes jump from 20 to 40 wt percentPl incorporate 25, 30 and 35 wt percent  data in each table alongwith changes in figures.

2. Conductivity of NP samples are quite good then why not applied in any electrochemical devices?

3. References must incorporated with new references

Author Response

We would like to thank the reviewer very much for the time spent analyzing our contribution. We appreciate your comments, and our answers are as follows.

(x) Extensive editing of English language and style required

We revised our manuscript for English and style, and all the corrections are marked up.

This paper is complete in the sense that it contains no systematic study i.e. synthesis, characterization and  correlation.

However I would like to suggest few modifications.

The answers to the referee’s specific comments are as follows.

  1. In composition most of tables takes a jump from 20 to 40 wt percentPl incorporate 25, 30 and 35 wt percent data in each table alongwith changes in figures.

We checked Our Figures and Tables for correctness, and we found the samples’ names displayed in the Figures are the same listed in the Tables. See the table below.

Analyzed Samples

GGLA-G0

GG-MMT1

GGLA-G1

GG-MMT5

GGLA-G2

GG-MMT10

GGLA-G5

GG-MMT15

GGLA-G10

GG-MMT20

GGLA-G20

GG-MMT40

GGLA-G40

Note that some Figures also display the results of the samples without clay, which was done to show the difference between SPEs and NPEs.

Nonetheless, we found that Tables 3, 4, and 9 had the bottom lines duplicated, i.e., the values were the same as in their first lines. Therefore, we erased the last lines in these Tables.

  1. Conductivity of NP samples are quite good then why not applied in any electrochemical devices?

We did apply in small ECD; however, the preliminary results could not be verified because of the pandemic and shout down of the labs. As we are back to our work since March, we will also start to test our membranes in ECDs.

  1. References must incorporated with new references

Three old references were erased, and we added four new references.

Erased references:

Fukushima, Y.; Inagaki, S., Synthesis of an intercalated compound of montmorillonite and 6-polyamide. In Inclusion Phenomena in Inorganic, Organic, and Organometallic Hosts, Springer: 1987; pp 365-374.

Armand, M. B., Polymer electrolytes. Annual Review of Materials Science 1986, 16, (1), 245-261, doi: 10.1146/annurev.ms.16.080186.001333.

Pandey, J. K.; Kumar, A. P.; Misra, M.; Mohanty, A. K.; Drzal, L. T.; Palsingh, R., Recent advances in biodegradable nanocomposites. J Nanosci Nanotechno 2005, 5, (4), 497-526, doi: 10.1166/jnn.2005.111.

Added references:

Osmałek, T.; Froelich, A.; Tasarek, S., Application of gellan gum in pharmacy and medicine. International Journal of Pharmaceutics 2014,  466, 328–340, 2014, doi: 10.1016/j.ijpharm.2014.03.038.

Sabadini, R. C.; Silva, M. M.; Pawlicka, A; Kanicki, J., Gellan Gum – O,O’-Bis(2-aminopropyl)-polyethylene glycol hydrogel for controlled fertilizer release. Journal of Applied Polymer Science 2018, 135, doi: 10.1002/app.45636.

Pacelli, S.; Paolicelli, P.; Moretti, G.; Petralito, S.; Di Giacomo, S.; Vitalone, A.; Casadei, M. A., Gellan gum methacrylate and laponite as an innovative nanocomposite hydrogel for biomedical applications. European Polymer Journal 2016, 77, 114–123, doi: 10.1016/j.eurpolymj.2016.02.007.

Bayzi Isfahani V. , P. R. F. P., Fernandes M.,  Sabadini R. C.,  Pereira S., Dizaji Hamid Rezagholipour , Arab A., Fortunato E., Pawlicka A., Rego R.,Zea Bermudez V. de , Silva M. M., Gellan-gum and LiTFSI-based solid polymer electrolytes for electrochromic devices. Chemistry Select 2021, 6, 5110–5119, doi: 10.1002/slct.202004614.

In addition to your comments, we improved Fig. 1 presentation by simplifying the inset temperatures display and by adding arrows to show the point outside the curvature. We also improved Fig. 5 presentation; we erased the inset label and put the sample names close to their results.

Reviewer 2 Report

This study examined how different amounts of montmorillonite (Na+SYN-1) clay addition affected the ionic conductivity of gellan gum-based nanocomposite polymer electrolytes (NPEs). The produced NPEs had undergone extensive characterization and showed good ionic conductivity. The work is well-presented, and the findings are intriguing. Therefore, after addressing the following concerns, I suggest that this manuscript be published.

1.     A brief description of gellan gum must be included in the introduction section. It should also incorporate previous studies on gellan gum-based electrolytes.

2.     (Page 3 Line 121) “In addition, we assembled ECDs containing GGLA/Na+SYN-1 electrolytes and featured them by cyclic voltammetry (CV), chronocoulometry, and UV-vis transmission spectroscopy”. The manuscript does not contain the data described above.

3.     How does the addition of different amounts of montmorillonite affect the morphology of the electrolyte samples? SEM images of the samples should be provided, and a discussion of how the morphology of the samples impacts the electrolyte properties.

4.     (Page 16 Line 479) Figure 9 is written as Figure 7.

Author Response

We would like to thank the reviewer very much for the time spent analyzing our contribution. We appreciate your comments, and our answers are as follows.

We revised our manuscript for English and style, and all the corrections are marked up.

This study examined how different amounts of montmorillonite (Na+SYN-1) clay addition affected the ionic conductivity of gellan gum-based nanocomposite polymer electrolytes (NPEs). The produced NPEs had undergone extensive characterization and showed good ionic conductivity. The work is well-presented, and the findings are intriguing. Therefore, after addressing the following concerns, I suggest that this manuscript be published.

Thank you very much.

A brief description of gellan gum must be included in the introduction section. It should also incorporate previous studies on gellan gum-based electrolytes.

We added a new paragraph with a brief description of gellan gum.

Gellan gum is a linear anionic exopolysaccharide consisting of -1,4-L-rhamnose, β-1,3-D-glucose, and β-1,4-D-glucuronic acid in a molar ratio of 1:2:1. Its native form has a high content of acyl, L-glyceryl, and acetyl groups. On the other hand, its deacetylated form has a low content of acyl groups, and this decreases its gelling and water absorption properties [28, 29]. Gellan gum has been investigated for several applications including biomedical [30], medical, and pharmaceutical [28], hydrogels for controlled fertilizer release [31] and soil humidity control [9], as well as solid polymeric electrolytes for electrochemical devices [32, 33], among others.

  1. (Page 3 Line 121) “In addition, we assembled ECDs containing GGLA/Na+SYN-1 electrolytes and featured them by cyclic voltammetry (CV), chronocoulometry, and UV-vis transmission spectroscopy”. The manuscript does not contain the data described above.

The referee is right. We erased this sentence as we do not show these results.

We characterized the nanocomposites by electrochemical impedance spectroscopy, thermal analyses of DSC and TGA, UV-vis reflectance, UV-vis transmittance, and EPR spectroscopies. In addition, we assembled ECDs containing GGLA/Na+SYN-1 electro-lytes and featured them by cyclic voltammetry (CV), chronocoulometry, and UV-vis transmission spectroscopy.

  1. How does the addition of different amounts of montmorillonite affect the morphology of the electrolyte samples? SEM images of the samples should be provided, and a discussion of how the morphology of the samples impacts the electrolyte properties.

The SEM images are a part of our second contribution related to this research, which will be soon submitted. However, we added one sentence at the end of the XRD discussion (line 287).

Finally, scanning electron microscopy (SEM) images (not shown here) also revealed a smoother surface of the sample GG-MMT40 when compared to GGLA-G40, due probably to the decrease of its crystallinity after the addition of clay.

  1. (Page 16 Line 479) Figure 9 is written as Figure 7.

It was corrected. Sorry about that.

In addition to your comments, we improved Fig. 1 presentation by simplifying the inset temperatures display and by adding arrows to show the point outside the curvature. We also improved Fig. 5 presentation; we erased the inset label and put the sample names close to their results.

Round 2

Reviewer 2 Report

The authors have adequately addressed all of the reviewer's concerns. Therefore, the manuscript can be accepted for publication